# *Streptococcus suis* Induces Macrophage M1 Polarization and Pyroptosis

**DOI:** 10.3390/microorganisms12091879

**Published:** 2024-09-12

**Authors:** Siqi Li, Tianfeng Chen, Kexin Gao, Yong-Bo Yang, Baojie Qi, Chunsheng Wang, Tongqing An, Xuehui Cai, Shujie Wang

**Affiliations:** 1State Key Laboratory for Animal Disease Control and Prevention, Harbin Veterinary Research Institute, Chinese Academy of Agricultural Sciences, Harbin 150001, Chinaantongqing@caas.cn (T.A.); 2College of Life Science, Northeast Forestry University, Harbin 150040, China; 3Heilongjiang Provincial Key Laboratory of Veterinary Immunology, Harbin 150069, China; 4Heilongjiang Research Center for Veterinary Biopharmaceutical Technology, Harbin 150069, China

**Keywords:** *Streptococcus suis*, inflammation, M1 macrophages, pyroptosis

## Abstract

*Streptococcus suis* is an important bacterial pathogen that affects the global pig industry. The immunosuppressive nature of *S. suis* infection is recognized, and our previous research has confirmed thymus atrophy with a large number of necrotic cells. In this current work, we aimed to uncover the role of pyroptosis in cellular necrosis in thymic cells of *S. suis*-infected mice. Confocal microscopy revealed that *S. suis* activated the M1 phenotype and primed pyroptosis in the macrophages of atrophied thymus. Live cell imaging further confirmed that *S. suis* could induce porcine alveolar macrophage (PAM) pyroptosis in vitro, displaying cell swelling and forming large bubbles on the plasma membrane. Meanwhile, the levels of p-p38, p-extracellular signal-regulated kinase (ERK) and protein kinase B (AKT) were increased, which indicated the mitogen-activated protein kinase (MAPK) and AKT pathways were also involved in the inflammation of *S. suis*-infected PAMs. Furthermore, RT-PCR revealed significant mRNA expression of pro-inflammatory mediators, including interleukin (IL)-1β, IL-6, IL-18, tumor necrosis factor (TNF)-α and chemokine CXCL8. The data indicated that the inflammation induced by *S. suis* was in parallel with pro-inflammatory activities of M1 macrophages, pyroptosis and MAPK and AKT pathways. Pyroptosis contributes to necrotic cells and thymocyte reduction in the atrophied thymus of mice.

## 1. Introduction

*Streptococcus suis* is an important global bacterial zoonotic pathogen [1,2] that causes meningitis, pneumonia, endocarditis, septicemia, arthritis and other symptoms in pigs, and it has caused considerable economic losses worldwide over the past 20 years [3,4]. At present, there are 29 serotypes based on the polysaccharide capsule [5]; of the 29 serotypes, the serotype 2 and serotype 14 strains are most often recoverable in human patients [6,7,8]. Although isolated cases of human infections with *S. suis* have been documented, a significant outbreak caused by *S. suis* serotype 2 occurred in Sichuan, China during the summer of 2005 [9]. A similar outbreak was identified in another province of China in 1998 [10,11]. Thus, *S. suis* is a serious threat to public health that has aroused widespread concern.

Macrophages play a crucial role as immune cells in various inflammatory and autoimmune diseases by participating in specific (cellular immunity) or non-specific defenses (innate immunity) within the body [12,13]. Their immune effects are highly flexible and depend on the current surrounding environmental stimuli [14]. According to their immunological function, macrophages can be differentiated into classically activated macrophages (M1) and alternately activated macrophages (M2) in the immune system, which have pro-inflammatory and anti-inflammatory functions, respectively [15]. M1 macrophages secrete pro-inflammatory cytokines and participate in positive immune responses, while M2 macrophages secrete anti-inflammatory cytokines and down-regulate immune responses [16,17].

Our previous work demonstrated thymic atrophy induced by *S. suis* infection, with large numbers of necrotic cells [10], which may be caused by programmed cell death pathways, including apoptosis, necroptosis and pyroptosis [6,18,19]. Pyroptosis is a type of inflammatory programmed cell death mode characterized by cell membrane pore formation, membrane rupture, cell swelling and release of cell contents [20]. Pyroptosis controls cytokine secretion, including IL-1β and IL-18, and contributes to antimicrobial immune defense [21]. Inflammatory caspase 1 cleaves the gasdermin D (GSDMD) protein to trigger pyroptosis. N-terminal fragment of GSDMD (GSDMD-N), which is the central executor of pyroptosis, promotes cell lysis by forming pores in the membranes [22].

The presence of pyroptosis and types of macrophage polarization found in the thymus during *S. suis* infection has not been reported. In this study, we aimed to determine the impact of *S. suis* on pyroptosis in the thymus of infected mice and specific cell populations. We show that *S. suis* triggered pyroptosis, M1 polarization, the activation of MAPK and AKT signaling pathways in macrophages and secretion of pro-inflammatory cytokines in the infected thymus. In addition, live cell imaging confirmed that *S. suis* infection can cause pyroptosis in primary porcine alveolar macrophages (PAMs) and immortalized PAMs (iPAMs) in vitro.

## 2. Materials and Methods

### 2.1. Bacterial Strains and Growth Conditions

The classical *S. suis* serotype 2 strain ATCC 700794 was stored in our lab and used throughout this study, cultured as previously described [23]. Briefly, *S. suis* 700794 was streaked and cultured on sheep blood agar plates (Thermo, Waltham, MA, USA) at 37 °C for 24 h, and individual colonies were inoculated in Todd Hewitt broth (THB) (BD, Franklin, TN, USA) and cultured in a shaking incubator at 37 °C for 18 h. The number of CFU/mL for broth cultures was determined by plating on Todd Hewitt agar (THA) in different dilutions and counting. Broth cultures were diluted in fresh culture medium before experiments based on the number of CFU/mL.

### 2.2. Animal Experiment

According to our previous studies [10], briefly, 20 six-week-old female C57BL/6 mice were inoculated by intraperitoneal injection (i.p.) with 0.25 mL of the bacterial suspension of *S. suis* serotype 2 700794 strain (5 × 10^7^ CFU) at 0 days post-inoculation (0 dpi), and 20 mice in a control group were inoculated with sterile THB. Mice were monitored daily for clinical signs and five infected mice and corresponding mice in the control group were euthanized humanely at 12 h post-inoculation (hpi) and 1, 2 and 4 dpi. Thymus samples were dissected from each mouse and were sectioned (6 μm-thick slices) on a cryostat and used for immunofluorescence staining. This mouse challenge experiment (approval number SY-2018-MI-012 and SY-2019-MI-058) was reviewed and approved by the Animal Experiment Ethics Committee of Harbin Veterinary Institute of the Chinese Academy of Agricultural Sciences and was conducted under the Guide for the Care and Use of Laboratory Animals of the Ministry of Science and Technology of China.

### 2.3. Cell and Culture

Primary porcine alveolar macrophages (PAMs) and iPAMs (PAM-Tang) were used for infection studies. To harvest primary PAMs, bronchoalveolar lavage was performed by injecting 50 mL sterile Roswell Park Memorial Institute-1640 (RPMI-1640, Gibco, Thermofisher, Waltham, MA, USA) containing 100 U/mL penicillin and 0.1 mg/mL streptomycin through trachea into major bronchi of both sides of the lungs. Lungs were massaged and bronchoalveolar lavage fluid was collected. The fluid was centrifuged at 800× *g* for 15 min to collect primary PAMs. The primary PAMs were washed twice with RPMI-1640 and counted. For the experiments, primary PAMs were further cultivated in RPMI-1640 supplemented with 10% (*v*/*v*) fetal bovine serum (FBS, Procell, Wuhan, China), 100 U/mL penicillin and 0.1 mg/mL streptomycin overnight at 37 °C with 5% CO_2_. iPAMs (PAM-Tang) were used in this study and cultured as described previously [6]. Briefly, iPAMs were cultured in RPMI-1640 medium (Gibco, Carlsbad, CA, USA) supplemented with 100 U/mL penicillin, 0.1 mg/mL streptomycin and 10% FBS (Procell, Wuhan, China) in a 5% CO_2_ incubator and passaged every 2 to 3 days to maintain subconfluence. For experiments, cell cultures were plated in laser confocal dishes or 12-well culture plates (NEST, Wuxi, China) at a density of 1 × 10^6^/well and 6-well culture plates (NEST, Wuxi, China) at a density of 2 × 10^6^/well.

### 2.4. LDH Release Assay

Lactate dehydrogenase (LDH) release assays were carried out as previously described [24], with some modifications. To determine the LDH enzyme release caused by *S. suis* 700794 infection, primary PAM cells were plated in 12-well culture plates (1 × 10^6^/well) and infected at a multiplicity of infection (MOI) of 1 with *S. suis* 700794 for 1, 2, 3, 8, 12 or 24 h, and then culture supernatant from infected primary PAMs at different time points (0 h, 1 h, 2 h, 3 h, 8 h, 12 h, 24 h) were collected and centrifuged for LDH enzyme detection using the CytoTox 96 Non-Radioactive Cytotoxicity assay kit (Promega, Madison, WI, USA) according to the manufacturer’s instructions. The absorbance of each well in 96-well plates at 492 nm wavelength was measured to determine the LDH activity.

### 2.5. Transmission Electron Microscopy

Primary PAM monolayers were grown on 20 mm Thermanox coverslips in a 6-well cell culture plate (2 × 10^6^/well), infected with *S. suis* 700794 (MOI = 1) for 1 h or 8 h. After two washes with PBS, the monolayers were fixed for 1 h at room temperature with cold 4% glutaraldehyde in 0.1 M PBS (pH 7.4) and further processed for electron microscopy as described previously [25]. Samples on coverslips were examined on a HITACHI H-7650 transmission electron microscopy (TEM).

### 2.6. Confocal Microscopy

Thymus samples collected were sectioned (8 μm-thick slices) and stored in our lab [10] and used for double-immunofluorescence staining. To determine the type of macrophage polarization in thymus tissue after bacterial infection, sections from infected mice (2 dpi) were stained with different antibodies, including rabbit anti-mouse CD206 (ABclonal, Beijing, China) or rabbit anti-mouse interleukin (IL)-1β (ABclonal). Macrophages that underwent pyroptosis in thymus tissue were stained with rabbit anti-mouse F4/80 antibody (Abcam, Cambridge, MA, USA) and rabbit anti-mouse GSDMD-N (Abcam) antibody. The specific process is shown in Table 1. Finally, sections were covered with the cover glass and observed in a laser-scanning confocal microscope.

### 2.7. RT-PCR

Total RNA was isolated from each thymus sample from 12 h post-infection (hpi), 1 day post-infection (dpi), 2 dpi and 4 dpi using the standard RNA Pre-Pure kit according to the producer (Tiangen, Beijing, China). RNA quality was examined with a Nanodrop spectrophotometer (Thermo, Waltham, MA, USA). Then, 5 μg of total RNA was reverse-transcribed into complementary (c) DNA using the RevertAid First Strand cDNA Synthesis Kit (Fermentas, Opelstrasse, Germany), followed by 50-fold dilution and 0.5 ng/μL (10 ng) for a reaction. SYBR^®^ Green qPCR Master Mix (Vazyme Biotech, Nanjing, China) was used to quantify the expression of inflammatory factors with specific primers [26] (Table 2). The expression of each inflammatory factor was normalized to β-actin and reported as fold changes.

### 2.8. Live-Cell Imaging

To examine the development of cell death in cultures infected with *S. suis* 700794, primary PAM and iPAM cells were first seeded into laser confocal dishes (1 × 10^6^ cells/well) with RPMI 1640 medium supplemented with 10% FBS overnight, and then the medium was replaced with RPMI 1640 with 3% FBS. Subsequently, primary PAM or iPAM cells were infected with *S. suis* 700794 (MOI = 1) and treated as indicated and stained with NucBlue™ Live Cell Stain (Hoechst 33342; Thermofisher) and propidium iodide (PI; BD Biosciences). Images were then captured using a laser-scanning confocal microscope (Carl Zeiss AG, Oberkochen, Germany).

### 2.9. Western Blotting

iPAMs were cultured in 6-well plates at a density of 2 × 10^6^ cells/well in RPMI 1640 supplemented with 10% FBS overnight. Cells were then infected with *S. suis* 700794 (MOI = 1) for 0, 15, 30, 60 or 120 min, and subsequently, cell samples were lysed in Radio Immunoprecipitation Assay (RIPA) lysis buffer (Beyotime, Shanghai, China) on ice for 10 min, containing the protease inhibitors 0.1 mM phenylmethylsulfonyl fluoride (PMSF) and Complete Protease Inhibitor (EDTA-free; Merck-Millipore, Darmstadt, Germany). BCA assay was used to determine the protein concentration of the samples. Equal amounts of protein were separated by 12.5% (*w*/*v*) sodium dodecyl sulfate-polyacrylamide gel electrophoresis and Spectra Multicolor Protein (Thermo Scientific, Waltham, MA, USA) as Ladder, then transferred to polyvinylidene fluoride (PVDF) membranes (MerckMillipore, Darmstadt, Germany). After blocking with 5% dry milk dissolved in PBS at room temperature for 1 h, membranes were incubated for 1 h at room temperature, targeting MAPK and AKT signaling pathways, including rabbit anti-AKT, p-38 and phospho-p38, ERK1/2 and phospho-ERK1/2, JNK1/2/3 and phospho-JNK1/2/3 (1:1000, Abcam) and anti-β-actin (1:200,000; ABclonal). After washing, HRP-conjugated secondary antibodies were added at 1:10,000 dilution for 1 h at room temperature, followed by washing with PBST buffer. HRP signals were detected with ECL-Supersensitive Luminescence Liquid (PerkinElmer Life Sciences, Fremont, CA, USA), followed by exposure to X-ray film. The Image v1.8.0 software (National Institutes of Health, Bethesda, MD, USA) was utilized for the quantification of the relative integral density of the target protein and the internal control. The relative integral density of the control was then normalized to enable a relative quantitative comparison.

### 2.10. Statistical Analysis

Data analysis was conducted using GraphPad Prism software (version 8.0.1; GraphPad Software Inc., La Jolla, CA, USA) and is expressed as the mean ± standard deviation (SD) of at least three replicates. The unpaired Student’s *t*-test was performed to calculate the *p*-values, with thresholds for statistical significance set at <0.05 (*), <0.01 (**) and <0.001 (***).

## 3. Results

### 3.1. S. suis Promotes M1 Macrophage Activation in Atrophied Thymus of Infected Mice

Macrophages regulate inflammatory responses by polarizing into different subtypes. To evaluate the subtypes of activated macrophages in the thymus of infected mice at 2 dpi, the expression levels of macrophage polarization markers IL-1β (M1) and CD206 (M2) were detected. The results showed that staining for the macrophage-specific antibody (F4/80) co-localized with IL-1β-labeled M1 macrophages, and expression of IL-1β in the infected thymus was increased (Figure 1). However, the expression of CD206 was almost absent, indicating that *S. suis* infection induced proinflammatory signals in the thymus of mice at the early stages of infection.

### 3.2. S. suis Primes Pyroptosis in Atrophied Thymus of Infected Mice

In order to further investigate pyroptosis in the *S. suis*-infected thymus of mice, the expression of GSDMD-N was evaluated in atrophied thymus tissue (Figure 2). The results showed that the expression of GSDMD-N in the thymus was clearly enhanced in mice at 2 days after infection by *S. suis*. Moreover, some signals from a macrophage-specific antibody (F4/80) co-localized with GSDMD-N fluorescence signals, which suggested that macrophages present pyroptosis in the thymus. These data indicate that *S. suis* could induce pyroptosis in the atrophied thymus of *S. suis*-infected mice and that the pyroptotic cells are macrophages.

### 3.3. High Expression of Proinflammatory Mediators in Atrophied Thymus of S. suis-Infected Mice

To understand the immune response triggered by *S. suis* in the thymus, the mRNA expression of several inflammatory mediators was measured. The results showed that *S. suis* induced significant expression of many inflammatory mediators. The expression of IL-6 was significantly increased at 12 hpi and 1 dpi and reached a peak at 12 hpi (Figure 3A) compared with the control group. The expression of CXCL8, IL-1β and IL-18 were significantly increased at 12 hpi, 1 dpi and 2 dpi, and reached a peak at 1 dpi compared with the control group (Figure 3B–D). Meanwhile, TNF-α was significantly increased and reached a peak at 1 dpi compared with the control group (Figure 3E). These data suggest that pro-inflammatory responses in the thymus were activated at the early stages of *S. suis* infection.

### 3.4. S. suis-Induced LDH Positively Correlates with the Infection Time In Vitro

To determine cytotoxicity in vitro, LDH release from infected primary PAMs was measured at 0, 1, 2, 3, 8, 12 and 24 hpi. The results showed that *S. suis* infection induced significant LDH activity in primary PAMs from 1 hpi to 24 hpi and positively correlated with the infection time (Figure 4A). TEM further confirmed that *S. suis* infection induced cytotoxicity and that the cytotoxicity of *S. suis* in primary PAM cells was time-dependent, with many vacuoles present at 8 hpi but not evident at 1 hpi (Figure 4B). These data demonstrated that the *S. suis*-treated primary PAMs released LDH and showed morphologic signs of cytotoxicity.

### 3.5. S. suis Induces Pyroptosis Both in iPAMs and Primary PAMs In Vitro

Live cell imaging was used to further confirm *S. suis*-induced pyroptosis in both iPAMs and primary PAMs in vitro. Primary PAM or iPAM cells were infected with *S. suis* 700794 (MOI = 1). At the morphological level, infected iPAMs exhibited increased pyroptotic membrane ballooning and took up propidium iodide (PI) red after *S. suis* infection for 8 h, compared with uninfected cells (Figure 5A; also see video, additional file 1). Primary PAMs infected with *S. suis* presented similar features after *S. suis* infection for 13 h, including cell swelling and large bubbles blowing from the plasma membrane. Subsequently, the cells were lysed and cellular contents were rapidly released (Figure 5B; also see video, additional file 2). Therefore, we concluded that *S. suis* induces pyroptosis in both iPAMs and primary PAMs in vitro.

### 3.6. S. suis Infection Actives the MAPK and AKT Signaling Pathways in PAMs

To investigate the effect of the classical inflammation-related MAPK signaling pathway on the *S. suis*-induced inflammatory response in PAMs, iPAMs were infected with *S. suis* 700794 (MOI = 1) and collected at 0, 15, 30, 60 or 120 min after infection for Western blot analysis. The results showed that *S. suis* upregulated p38 and ERK phosphorylation in infected iPAMs compared with the control group. As shown in Figure 6A,B, the levels of p38 and ERK phosphorylation increased significantly, peaked at 15 min after infection and finally returned to control levels, while JNK phosphorylation had no significant change, which indicated that *S. suis* regulated the inflammatory response by activating the p38 and ERK pathways rather than the JNK pathway. Also, the AKT protein increased from 15 min to 120 min after infection and reached a peak at 30 min after infection (Figure 6A,F), which indicated that *S. suis* infection activated the AKT pathway. Altogether, the MAPK and AKT signaling pathways are involved in proinflammatory responses in *S. suis*-infected PAMs.

## 4. Discussion

The thymus is an important central immune organ in mammals [27], and we previously showed that *S. suis* infection results in significant necrosis and reduced lymphocytes within the thymus tissue as early as 1 dpi [10]. Since pyroptosis presents characteristics very similar to those of necrosis, it was unknown whether pyroptosis was present in macrophages within the thymus tissue after *S. suis* infection. Here, pyroptosis on macrophages in the atrophied thymus of *S. suis*-infected mice was identified with confocal microscopy, and pyroptosis in primary PAMs and iPAMs after *S. suis*-infection has also been demonstrated in vitro by live cell imaging.

Pyroptosis is a form of gasdermin-mediated programmed necrosis and a common mechanism of innate immune effects, which stimulates inflammatory responses [28]. Pyroptosis enhances immune defense function by destroying the protective intracellular replication niche of pathogens [29]. Moderate pyroptosis is a host defense against infectious pathogens, but hyperactive pyroptosis may be associated with intemperate inflammatory responses and serious tissue damage [20]. This study confirmed that pyroptosis happened in macrophages in the atrophied thymus of *S. suis*-infected mice, which indicated the presence of pyroptosis in the necrosis cells of the atrophied thymus found in a previous study. Meanwhile, we also confirmed pyroptosis in macrophages following *S. suis* infection in vitro, with cell swelling and large bubbles blowing from the plasma membrane and increased dying cells (PI+), resulting in the release of IL-1β and IL-18. In this study, the mRNA expression levels of IL-1β and IL-18 indeed increased significantly in the atrophied thymus, in which the existence of pyroptosis was further confirmed. In addition, released LDH was used to measure cytotoxicity and would also be released by cells undergoing pyroptosis in response to inflammasome activation. This study has confirmed that *S. suis* is capable of inducing LDH release from primary PAMs in vitro, and this effect is observed to increase with prolonged culture time, which is associated with pyroptosis and consistent with previous studies [30].

Macrophages are an important part of the thymus cells and regulate inflammatory responses, and can differentiate into specific phenotypes through polarization, which permits specific biological functions in response to microenvironmental stimuli [31]. The fluorescence imaging showed *S. suis* infection activated primarily M1-type macrophages as early as 1 dpi in *S. suis*-infected thymus. The activation of the M1 macrophage occurs in an inflammation-dominated environment, in turn promoting the production of a large number of proinflammatory cytokines, including IL-1, IL-6, IL-18, tumor necrosis factor-α (TNF-α) and other chemokines [32]. *S. suis* infection in a previous study induced an abnormal serum cytokine dynamic, including the levels of serum IL-2, IL-4, IL-6 and TNF-β, which increased significantly [10]. Also, the proinflammatory mediators (IL-1β, IL-6, IL-18, TNF-α and CXCL8) were found to be highly expressed in the thymus tissue from *S. suis*-infected mice in this study. MAPK signaling can regulate inflammatory responses, its related signaling kinases including JNK, ERK and p38 [33]. The up-regulating of the p38 and ERK signaling pathways can inhibit M2-type polarization and promote M1-type polarization of macrophages [34]. The results of this study showed exactly that the p38 and ERK signaling pathways were up-regulated and M1 macrophage polarization occurred in *S. suis*-infected thymus tissue, which indicated that the polarization of macrophages is closely related to the activation of the p38 and ERK signaling pathways in MAPK signaling pathways.

## 5. Conclusions

This study demonstrated that *S. suis* could activate M1 polarization, the MAPK signaling pathway and prime pyroptosis in macrophages, as well as produce inflammatory mediators and induce programmed cell death, which contribute to the reduction of lymphocytes in the atrophied thymus of infected mice. However, the underlying mechanism of pyroptosis in macrophages is undetermined and warrants further investigation.

## Figures and Tables

**Figure 1 microorganisms-12-01879-f001:**
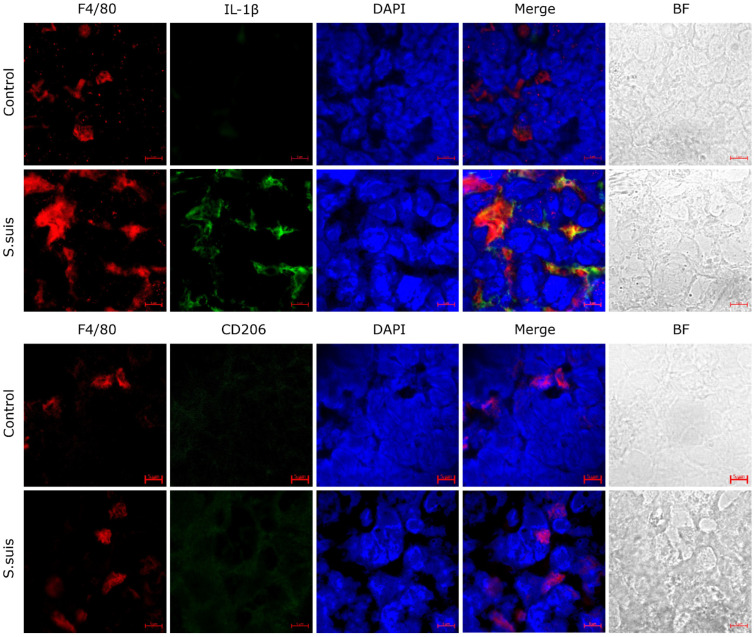
Thymic macrophage polarization in *S. suis* infection. Using confocal laser scanning microscopy, appropriate FITC-conjugated antibodies were used to label cells in sections from thymus of infected mice; M1 macrophage marker IL-1β (green); M2 macrophage marker CD206 (green). Macrophages were stained with F4/80 antibody (red) and cell nuclei (blue) were stained with DAPI; BF: Bright-field. Scale bars, 5 μm.

**Figure 2 microorganisms-12-01879-f002:**
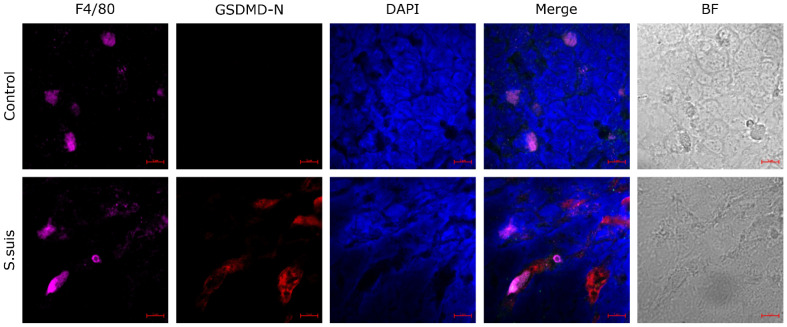
*S. suis* induces pyroptosis in macrophages of the thymus of infected mice. Confocal laser scanning immunostaining was performed to visualize thymic macrophages (F4/80 antibody; purple) and pyroptosis (GSDMD-N; red) in atrophied and control thymus of mice, and nuclei were stained with DAPI (blue). BF: Bright-field. Scale bars, 5 μm.

**Figure 3 microorganisms-12-01879-f003:**
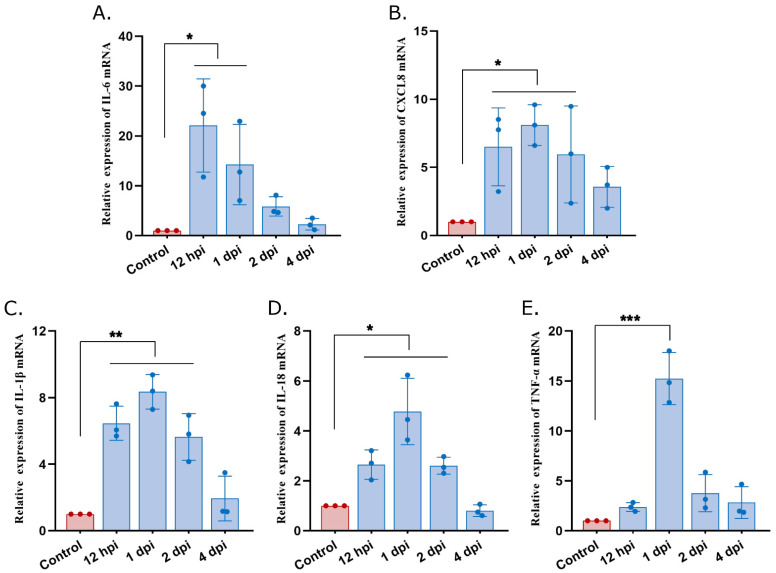
*S. suis* induces pro-inflammatory mediators in thymus of infected mice. (**A**) Total tissue RNA was extracted at 12 hpi and 1, 2 and 4 dpi, and the levels of IL-6, (**B**) CXCL8, (**C**) IL-1β, (**D**) IL-18, (**E**) and TNF-α mRNA expression were analyzed using real-time RT-PCR. Experiments were repeated three times. Differences were analyzed using unpaired Student’s *t*-test; data are mean ± SD; *: *p* < 0.05; **: *p* < 0.01; ***: *p* < 0.001.

**Figure 4 microorganisms-12-01879-f004:**
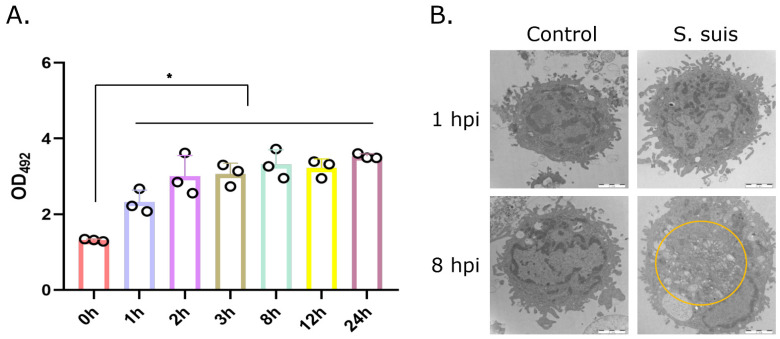
*S. suis*-induced LDH release and damage to primary PAMs. Primary PAMs were infected with *S. suis* 700794 at an MOI = 1 for 1, 2, 3, 8, 12 or 24 h, with a negative control included as 0 h. (**A**) LDH release induced by *S. suis* in primary PAMs was quantified by a standard LDH-release assay. (**B**) Electron micrographs of infected primary PAMs showing *S. suis*-induced cellular damage at 8 hpi, and yellow circle showed vacuoles of PAMs. Experiments were repeated three times, and results are expressed as means ± S.D., as determined by one-way ANOVA; *: *p* < 0.05; scale bars, 2 μm.

**Figure 5 microorganisms-12-01879-f005:**
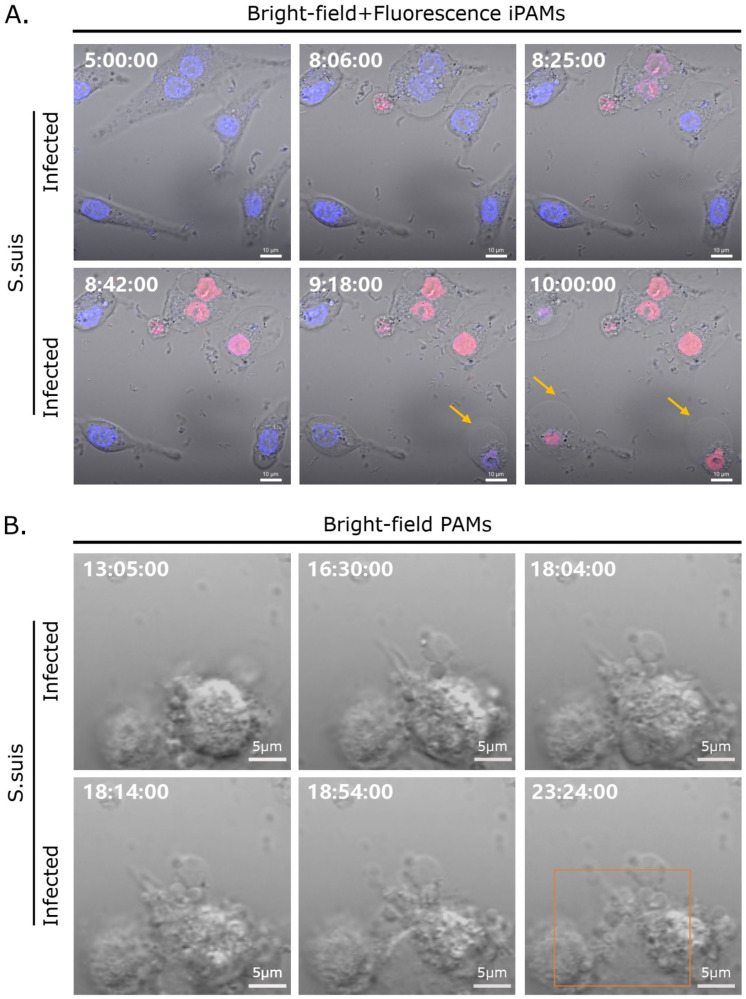
Live-cell imaging of pyroptosis after in vitro infection with *S. suis*. PAMs were cultured in glass-bottom cell culture dishes and treated with *S. suis* (MOI = 1). (**A**) Microscopy images of iPAMs treated with *S. suis* incubated in medium containing Hoechst 33342 (cell nuclei; blue) and propidium iodide (PI; red), the yellow arrows indicate pyrolyzed cells. (**B**) Microscopy images of primary PAMs treated with *S. suis*, and rectangle indicates pyrolyzed cells. Numbers at upper left indicate time post-infection (h:min:s).

**Figure 6 microorganisms-12-01879-f006:**
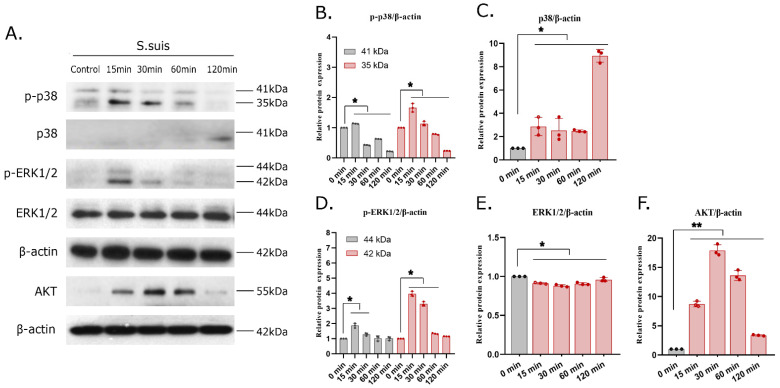
The MAPK and AKT pathways were activated by *S. suis* in porcine macrophages. Cells were infected with *S. suis* at an MOI = 1 for 15, 30, 60 and 120 min. The cell lysates were collected, and levels of proteins related to inflammatory functional indexes were determined by Western blot (**A**). Representative Western blots show the expression level of (**B**) p-p38, (**C**) p38, (**D**) p-ERK1/2, (**E**) ERK1/2 and (**F**) AKT. The mean ± SD of experiments are shown; *: *p* < 0.05; **: *p* < 0.01.

**Table 1 microorganisms-12-01879-t001:** Protocols for confocal microscopy.

Cell Surface	Antibody	Dilution, Temperature, Time	Fluorescent Secondary Antibody Temperature, Time
M1 Macrophage	IL-1β	1:200, overnight, 4 °C	Alexa Fluor 488-conjugated goat anti-mouse IgG, RT, 1 h
M2 Macrophage	CD206	1:100, overnight, 4 °C	Alexa Fluor 488-conjugated goat anti-mouse IgG, RT, 1 h
Macrophage	F4/80	1:50, overnight, 4 °C	Alexa Fluor 647-conjugated rabbit anti-mouse F4/80, RT, 1 h
Pyroptosis	GSDMD-N	1:50, overnight, 4 °C	Alexa Fluor 568-conjugated goat anti-rabbit antibody, RT, 1 h
Nuclei	DAPI	RT, 20 min	

**Table 2 microorganisms-12-01879-t002:** Mouse primer pairs used for real-time PCR.

Gene	Forward Primer (5′-3′)	Reverse Primer (5′-3′)
IL-6	TGTGCAATGGCAATTCTGAT	GGTACTCCAGAAGACCAGAGGA
IL-8	GGTCTTCCTGCTTGAATGGCTTGA	GCGGTGTCCTGATTATCGTCCTC
IL-18	GCCATGTCAGAAGACTCTTGCGTC	GTACAGTGAAGTCGGCCAAAGTTGTC
IL-1β	GGTGTGTGACGTTCCCATTAGAC	AGGGTGGGTGTGCCGTCTT
TNF-α	GTCTACTGAACTTCGGGGTGATCG	TGGGCTACAGGCTTGTCACTCG
β-actin	GCTTCTTTGCAGCTCCTTCGT	CCTTCTGACCCATTCCCACC

## Data Availability

The original contributions presented in the study are included in the article, further inquiries can be directed to the corresponding authors.

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
