# Peer review of "Streptococcus suis Induces Macrophage M1 Polarization and Pyroptosis"

_microorganisms, 2024, doi:10.3390/microorganisms12091879_

Round 1

Reviewer 1 Report

Comments and Suggestions for Authors

In this manuscript, the authors aim to demonstrate that S. suis triggers pyroptosis, M1 polarization, activation of MAPK and AKT signaling pathways in macrophages, and secretion of pro-inflammatory cytokines in the infected thymus. They also report that S. suis infection can cause pyroptosis in primary porcine alveolar macrophages and immortalized PAMs in vitro.

The authors report a lot of data, but the manuscript needs improvement

Abstract

According to the journal, the abstract must be 200 words. Please reduce it.

Line 14: S. suis is not a disease, but it can cause disease

Introduction

Lines 54-59: Please divide the sentence where you talk about IL1 and 18 from the sentence where you talk about gasdermin. Moreover, specify the role of caspase 1 in the cleavage process of gasdermin.

Material and Methods:

line 71: Did you use an ATCC reference strain? Please, specify if 700794 is the ATCC number

line 95: To collect the primary PAMs, how much RPMI did you use?

lines 130-143: I suggest schematizing these protocols in a table

line 146: according to the producer?

line 149: How much RNA was put into the reaction? What concentration?

Table 1: Were the primers used drawn on this occasion? If so, it would be appropriate to specify the region of interest for each gene and the melting temperature. If not, it is necessary to add the reference

line 172: At what concentration of polyacrylamide were the gels you used? What kind of Ladder did you use? Please specify

Results

lines 189-191, 205-207, 220-221, 278-280: These sentences are not results. They should be moved to the introduction

lines229-231, 246-248, 267-268: These sentences are not results. They should be moved to the discussion

line 195: "the infected thymus was significantly increased"  I suggest deleting the word significantly

3.1. S. suis promotes M1 macrophage activation in atrophied thymus of infected mice: This is data obtained how many hours after infection? Specify

line 212-213: these are conclusions

The discussion deserves to be improved

Author Response

August 22, 2024

Re. Ms. No. 3142806

Title: Streptococcus suis Induces Macrophage M1 Polarization and Pyroptosis in Atrophied Thymus and in vitro

Microorganisms

Dear reviewer 1,

We sincerely thank you for the constructive criticisms and valuable comments, which were of great help in revising the manuscript (Streptococcus suis Induces Macrophage M1 Polarization and Pyroptosis in Atrophied Thymus and in vitro). We have revised the manuscript based on the comments of you. We believe the revised manuscript has been significantly improved, and our responses to your comments are given below and highlighted in the manuscript.

Reviewer 1

In this manuscript, the authors aim to demonstrate that S. suis triggers pyroptosis, M1 polarization, activation of MAPK and AKT signaling pathways in macrophages, and secretion of pro-inflammatory cytokines in the infected thymus. They also report that S. suis infection can cause pyroptosis in primary porcine alveolar macrophages and immortalized PAMs in vitro. The authors report a lot of data, but the manuscript needs improvement.

  1. Abstract

According to the journal, the abstract must be 200 words. Please reduce it.

Line 14: S. suis is not a disease, but it can cause disease

------ Thanks for this suggestion. The abstract has been revised according to the reviewer’s suggestion.

  1. Introduction

Lines 54-59: Please divide the sentence where you talk about IL1 and 18 from the sentence where you talk about gasdermin. Moreover, specify the role of caspase 1 in the cleavage process of gasdermin.

------ Thanks for this comment. The introduction has been revised according to the reviewer’s suggestion.

  1. Material and Methods:

line 71: Did you use an ATCC reference strain? Please, specify if 700794 is the ATCC number

line 95: To collect the primary PAMs, how much RPMI did you use?

lines 130-143: I suggest schematizing these protocols in a table

line 146: according to the producer?

line 149: How much RNA was put into the reaction? What concentration?

------ We appreciate for the reviewer’s comments and apologized for the incomplete introduction. Material and Methods has been supplemented in the lines 88, 113, 150-154, 159-166 of the revised manuscript according to the reviewer’s suggestion.

  1. Table 1: Were the primers used drawn on this occasion? If so, it would be appropriate to specify the region of interest for each gene and the melting temperature. If not, it is necessary to add the reference

------ We appreciate this comment and apologized for the incomplete reference about the table used in this study. We have added the reference of the primers used in the line 165 of the revised manuscript.

  1. line 172: At what concentration of polyacrylamide were the gels you used? What kind of Ladder did you use? Please specify

------ Thanks for this comment. The concentration of polyacrylamide and kind of Ladder have been added in lines 184-186 of the revised manuscript.

  1. Results

lines 189-191, 205-207, 220-221, 278-280: These sentences are not results. They should be moved to the introduction.  Lines 229-231, 246-248, 267-268: These sentences are not results. They should be moved to the discussion. line 195: "the infected thymus was significantly increased"  I suggest deleting the word significantly. 3.1. S. suis promotes M1 macrophage activation in atrophied thymus of infected mice: This is data obtained how many hours after infection? Specify. line 212-213: these are conclusions

------ We appreciate these comments and have revised manuscript according to the reviewer’s suggestions.

  1. The discussion deserves to be improved

------ Thanks for this suggestion. The discussion has been improved in the revised manuscript.

We have addressed the comments and concerns from you and have found them instructive and helpful for improving our manuscript. We hope the revised manuscript will be suitable for publication in Microorganisms.

Yours sincerely,

Shujie Wang

Reviewer 2 Report

Comments and Suggestions for Authors

Streptococcus suis is a bacterium associated with diseases affecting the swine industry. It is also a zoonotic bacterium. The immunosuppressive nature of S. suis infection has been previously demonstrated, including the occurrence of thymic atrophy with the presence of large numbers of necrotic cells.

The study described here by Li et al. investigated the role of pyroptosis in cellular necrosis in specific thymic cell populations in S. suis infected mice. Confocal microscopy revealed that S. suis infection activated M1 macrophages with pyroptosis in atrophied thymic macrophages. Live cell imaging further confirmed that S. suis may be involved in pyroptosis.

In my opinion, this study is very important. According to the manuscript, the authors seem to have performed a good experimental job. The methodology was well described and the results section present very interesting findings, well demonstrated in figures and images. Therefore, the study should be published.

However, I do not think the manuscript is ready to be published in Microorganisms. And some modifications are necessary previously to a deeper per review. First of all I suggest shortening the title of the article (Streptococcus suis Induces Macrophage M1 Polarization and Pyroptosis). Second, the Abstract must be improved to present more clearly the main findings. Third, the Introduction is not presenting some more detailed information of the microorganism taxonomy and its epidemiological importance worldwide. I believe that the epidemiological data are very relevant in the study and should be introduced in the first paragraphs. Also the Discussion section should be improved, revising deeper the scientific literature. It is also necessary to describe the limitations of the study as well as the importance for further studies. Therefore, I think authors need to review all the manuscript and prepare it better to be peer reviewed again.

Finally, careful proofreading of the English language is also required, checking duplicity (just one example: ... disease... disease in the first sentence of the Abstract), correct scientific nomenclature of bacteria, text for grammar, style and syntax. I also recommend avoiding the use of the first-person plural pronouns (“we”, “our”, etc.). Please choose passive voice.

Comments on the Quality of English Language

Moderate editing of English language required.

Author Response

August 22, 2024

Re. Ms. No. 3142806

Title: Streptococcus suis Induces Macrophage M1 Polarization and Pyroptosis in Atrophied Thymus and in vitro

Microorganisms

Dear reviewer 2,

We sincerely thank you for the constructive criticisms and valuable comments, which were of great help in revising the manuscript (Streptococcus suis Induces Macrophage M1 Polarization and Pyroptosis in Atrophied Thymus and in vitro). We have revised the manuscript based on the comments of you. We believe the revised manuscript has been significantly improved, and our responses to your comments are given below and highlighted in the manuscript.

Streptococcus suis is a bacterium associated with diseases affecting the swine industry. It is also a zoonotic bacterium. The immunosuppressive nature of S. suis infection has been previously demonstrated, including the occurrence of thymic atrophy with the presence of large numbers of necrotic cells. The study described here by Li et al. investigated the role of pyroptosis in cellular necrosis in specific thymic cell populations in S. suis infected mice. Confocal microscopy revealed that S. suis infection activated M1 macrophages with pyroptosis in atrophied thymic macrophages. Live cell imaging further confirmed that S. suis may be involved in pyroptosis. In my opinion, this study is very important. According to the manuscript, the authors seem to have performed a good experimental job. The methodology was well described and the results section present very interesting findings, well demonstrated in figures and images. Therefore, the study should be published. However, I do not think the manuscript is ready to be published in Microorganisms. And some modifications are necessary previously to a deeper per review.

  1. First of all I suggest shortening the title of the article (Streptococcus suis Induces Macrophage M1 Polarization and Pyroptosis).

------ Thanks for this suggestion. The title has been shortened according to the reviewer’s suggestion.

  1. Second, the Abstract must be improved to present more clearly the main findings.

------ Thanks for this suggestion. The abstract has been revised according to the reviewer’s suggestion.

  1. Third, the Introduction is not presenting some more detailed information of the microorganism taxonomy and its epidemiological importance worldwide. I believe that the epidemiological data are very relevant in the study and should be introduced in the first paragraphs.

------We thank very much for the comments. The introduction has been revised according to the reviewer’s suggestion.

  1. Also the Discussion section should be improved, revising deeper the scientific literature. It is also necessary to describe the limitations of the study as well as the importance for further studies. Therefore, I think authors need to review all the manuscript and prepare it better to be peer reviewed again.

------ Thanks for this suggestion. The discussion has been improved in the revised manuscript.

  1. Finally, careful proofreading of the English language is also required, checking duplicity (just one example: ... disease... disease in the first sentence of the Abstract), correct scientific nomenclature of bacteria, text for grammar, style and syntax. I also recommend avoiding the use of the first-person plural pronouns (“we”, “our”, etc.). Please choose passive voice.

 ------Thanks for your suggestions and we have revised and double checked the manuscript to avoid the minor errors.

We have addressed the comments and concerns from you and have found them instructive and helpful for improving our manuscript. We hope the revised manuscript will be suitable for publication in Microorganisms.

Yours sincerely,

Shujie Wang

Round 2

Reviewer 1 Report

Comments and Suggestions for Authors

The authors have addressed all points and have revised the manuscript to publication standards. 

Author Response

Dear reviewer 1,

Thank you very much!

Best,

Shujie Wang